# Electroactive Polymer-Based Composites for Artificial Muscle-like Actuators: A Review

**DOI:** 10.3390/nano12132272

**Published:** 2022-07-01

**Authors:** Aleksey V. Maksimkin, Tarek Dayyoub, Dmitry V. Telyshev, Alexander Yu. Gerasimenko

**Affiliations:** 1Institute for Bionic Technologies and Engineering, I.M. Sechenov First Moscow State Medical University, Bolshaya Pirogovskaya Street 2-4, 119991 Moscow, Russia; telyshev@bms.zone (D.V.T.); gerasimenko@bms.zone (A.Y.G.); 2Institute of Biomedical Systems, National Research University of Electronic Technology, 124498 Moscow, Russia

**Keywords:** electroactive polymers (EAPs), artificial muscles, dielectric EAPs, ionic gels, electrostrictive graft elastomers, liquid crystal elastomers, ionic gels, conductive polymers, ionic polymer-metal composites

## Abstract

Unlike traditional actuators, such as piezoelectric ceramic or metallic actuators, polymer actuators are currently attracting more interest in biomedicine due to their unique properties, such as light weight, easy processing, biodegradability, fast response, large active strains, and good mechanical properties. They can be actuated under external stimuli, such as chemical (pH changes), electric, humidity, light, temperature, and magnetic field. Electroactive polymers (EAPs), called ‘artificial muscles’, can be activated by an electric stimulus, and fixed into a temporary shape. Restoring their permanent shape after the release of an electrical field, electroactive polymer is considered the most attractive actuator type because of its high suitability for prosthetics and soft robotics applications. However, robust control, modeling non-linear behavior, and scalable fabrication are considered the most critical challenges for applying the soft robotic systems in real conditions. Researchers from around the world investigate the scientific and engineering foundations of polymer actuators, especially the principles of their work, for the purpose of a better control of their capability and durability. The activation method of actuators and the realization of required mechanical properties are the main restrictions on using actuators in real applications. The latest highlights, operating principles, perspectives, and challenges of electroactive materials (EAPs) such as dielectric EAPs, ferroelectric polymers, electrostrictive graft elastomers, liquid crystal elastomers, ionic gels, and ionic polymer–metal composites are reviewed in this article.

## 1. Introduction

Actuator refers generally to a device that converts applied energy to motion. In present times, in contrast to the traditional heavy and complicated actuators, such as metallic and ceramic ones which consist of motors and associated mechanisms, polymeric actuators are more adopted in biological systems because of their prospective properties, such as light weight, simple, quiet, biodegradable, and fast response, in addition to their good mechanical properties, especially high strain [1,2]. Being the most important, artificial muscle is considered the main application of polymeric actuators [3,4]. Under the impact of external stimuli (Figure 1), such as chemical stimulus (pH changes), electric, humidity, light, temperature, magnetic field, etc., the controllable changes can be in size or shape. However, by removing the external effects, actuatable materials can recover their initial shapes and sizes.

One of the main types of actuators is actuators based on electroactive polymers (EAPs) [5,6]. EAPs are polymers that change their mechanical or optical characteristics, by the application of an electrical field, with accurate control of energy transformation from electrical to mechanical and vice versa. These characteristics, such as good actuation strains, high energy, good scalability, and low cost, make EAPs highly attractive to be used in applications such as sensors and artificial muscle actuators [7,8]. As can be seen in Figure 2, electroactive polymers can be divided into two groups: dielectric (electronic) and ionic EAPs [9,10,11,12,13,14,15]. Dielectric EAPs contain dielectric elastomers, ferroelectric polymers, electrostrictive graft elastomers, and liquid crystal elastomers. On the other hand, ionic EAPs can be divided into ionic polymer gels, conductive polymers, and ionic polymer–metallic composites [14]. In general, the electric field generated on the surface of polymers is the main driving force for electroactive polymers. Under a stimulus of an electric field, dielectric EAPs are driven by the accumulation and interaction of the opposite electric charges on the surface of elastomers, while ionic EAPs are activated by the mobility or the diffusion of ions.

## 2. Dielectric Electroactive Polymers

### 2.1. Dielectric Elastomers

A dielectric elastomer (DE) actuator consists of a thin and soft elastomer film placed between two compliant electrodes, Figure 3. The construction of the DE actuator is similar to a flexible capacitor. When connected to a source of direct current (DC), compliant electrodes are charged with opposite charges; therefore, the accumulated opposite charges on the electrodes are attracted to each other via the Coulomb force, which leads to the formation of an electrostatic pressure (P). Since the elastomer films have a low Young’s modulus (0.002–10 MPa), the formed electrostatic pressure leads to an easy deformation of the elastomer film in the plane (thickness becoming thinner), Figure 3. Due to their elastic properties, the original thickness of the elastomer film is restored when the DC voltage is released. This is the general operating principle of the DE actuators by converting electrical to mechanical energy.

The power of the DE actuators depends on the electrostatic pressure, which is determined by Equation (1). The magnitude of the generated electrostatic pressure (P) proportionally increases by raising the DC voltage (U), applied on the compliant electrodes, and the relative permittivity (ԑ_r_) of the film, or by decreasing the elastomer film thickness (d). The relative permittivity of the dielectric elastomer film is considered a very important parameter to achieve a highly powerful DE actuator. It characterizes the ability of the material to polarize under the influence of an external electric field, which means that the dielectric material with a higher relative permittivity can generate a higher polarization degree. In the initial state (DC voltage off), the molecular dipoles of elastomer films are in a disordered state, Figure 4. Under the external electric field (purple lines, Figure 4), the elastomer film will be polarized (i.e., the molecular dipoles will be oriented), in which their positive charges will be displaced in the electric field direction, while the negative ones will be shifted in the opposite field direction. In the case of weakly bonded molecules of the elastomer, these molecules can reorient in a way that their symmetry axes are rearranged with the electric field direction [17]. Polarized molecular dipoles can create an internal electric field (red lines, Figure 4), which leads to the attenuation of the external electric field passing through the elastomer film. This effect will lead to a reduction of the interaction among the opposite charges accumulated on the DE actuator electrodes, i.e., although the potential difference among the electrodes decreases, the current source will keep it increasing until it reaches the initial voltage value (U) because of continuous charging. The polarization of the elastomer film allows the accumulation of more opposite charges on the surface of the electrodes because the charges of the molecular dipoles on the surface of the elastomer film are compensated by the charges accumulated on the electrodes. This effect leads to an increase in the (P) electrostatic pressure, which causes the deformation of the elastomer film.
P = ԑ_0_ × ԑ_r_ × (U/d)^2^
(1)
where P—electrostatic pressure, ԑ_0_—free space permittivity (8.85 × 10^−12^ F/m), ԑ_r_—relative permittivity of the dielectric elastomer film, U—DC voltage, and d—thickness of the dielectric elastomer film.

The increase of the DC voltage on the electrodes is proportional to the stored energy in the DE actuator. The maximum value of the DC voltage is limited by the dielectric strength of the elastomer film. In-plane pre-stretching of the elastomer is considered an effective method to increase the dielectric strength [18,19,20] and reduce the thickness of the elastomer film.

DE actuators can have different configurations of deformation, such as spiral, cone, reel, and rhombus, which are related to the generated types of movements: compression; elongation; bending; and rotation; Figure 5.

The choice of dielectric material is a key factor that determines the performance of the actuators based on them. Siloxanes, acrylates, and polyurethanes are known dielectric elastomers having low Yong’s modulus, high deformation strain, and dielectric strength, which make them applicable for the creation of high-performance DE actuators [21]. However, the low relative permittivity of the above-mentioned elastomers requests using a high driving DC voltage (>1 kV) for actuation of prepared actuators based on them. The need to use DC voltage higher than 1 kV requires the application of additional specific semiconductor devices, which leads to more complications to activate and control DE actuators. High voltage can harm the human tissues and destroy the device employed, which limits the usage of dielectric materials in medical applications [22,23,24,25,26]. Therefore, to apply them in the medical field requires the reduction of driving voltage. This can be achieved by increasing the relative permittivity (ε_r_) of dielectric elastomers by using two main ways: incorporation of conductive high relative permittivity filler to the elastomer matrix and chemical modifications of the elastomer.

In reference [27], a polymer of vinyl chloride and ethylene copolymer (P(VC-E)) was prepared using assisted reduction of PVC by TTMSS (Tris(trimethylsilyl)silane). The authors found that P(VC-E) with a 40 mol% of VC had the highest breakdown strength (85 MVm^−1^) and a high relative permittivity (>10) and under a low voltage (450 V), it presented a high cyclical operability with a lifelong time. Moreover, they mentioned that the enhanced electromechanical performance of their prepared composite, since as the Maxwell stress was considered the only source of deformation under the electric field, it was related to its high permittivity at room temperature, high modulus, and breakdown strength caused by the strain-stiffening effect. The two elements, high permittivity and the large values of the Maxwell stress, were related to the polar C-Cl bonds (Figure 6).

In the reference [28], they chemically grafted hydroxy-terminated polydimethylsiloxane (OH-PDMS) to the carbon black (CB) surface using isophorone diisocyanate as a crosslinking agent to enhance the CB chemical compatibility and its insulation property, Figure 7. The authors showed that encapsulated CB particles exhibited a lower aggregation compared to normal CB, which led to a higher polarized capacitance.

In general, the pros of the actuators based on dielectric elastomer EAPs are their large displacements of strain up to 200–380%, high energy density (3.4 J/g) [15,29], fast response (maximum contraction frequency can reach 1000 Hz) [30], and being considered cheap. On the other hand, the cons of these polymers are their high driving voltage (~150 MV/m) and the occurrence of pre-strain and compromise in the actuation force due to their high displacements.

### 2.2. Ferroelectric Polymers

Ferroelectricity means the occurrence of a spontaneous electric polarization in a dielectric or non-conducting material, which can be reversed by the application of an external electric field [31]. This phenomenon is called piezoelectricity. In piezoelectric material, the charges of the molecular dipoles are displaced and therefore uncompensated. As a result of the polarization of the aligned molecular dipoles and uncompensated charges, the piezoelectricity generates an internal electric field. The internal electric field attracts charges to the piezoelectric surface, which forms an external electric field directed at balancing the internal electric field. In the initial state, both electric fields are compensated, Figure 8. Furthermore, electric current will not flow unless a mechanical stress is applied on the piezoelectric material. The direction of the electric current flow (I) will depend on the type of applied load (compressive or tensile) to the piezoelectric material. When a tensile load is applied, the dipole moment (P_s_) of the piezoelectric material increases, which increases the internal electric field. Potential difference between the surfaces of piezoelectric material occurs and neutralizing current flow (I) flows to the direction opposite to the dipole moment (P_s_) (electrons move from one electrode to the opposite to increase the external electric field). When a compressive load is applied, the dipole moment (P_s_) of the piezoelectric material decreases, which decreases the internal electric field. In this case, the neutralizing current flow flows in the same direction as to the dipole moment (P_s_) to balance the surface charge. After removal of mechanical stresses, the piezoelectric material recovers its original shape due to elastic properties. As can be seen in Figure 8b, by applying an electric field, the piezoelectric material will deform and the deformation type (compression or expansion) will depend on the electric field direction [32,33].

Poly(vinylidene fluoride) (PVDF) and its copolymers and polypropylene (PP) are the most famous ferroelectric polymers [34,35,36]. Piezoelectric polymers can be divided into several groups depending on their topology and dipole moment, Figure 9. The first group is bulk polymers, which consist of amorphous and semi-crystalline polymers. The piezoelectric properties of this group depend on their molecular structure and orientation [10,37]. Nevertheless, the molecular structure of the polymer shall have molecular dipoles, which can be reoriented within the polymer and kept in their preferred orientation state. For semi-crystalline polymers, by applying stress, the polar groups of the polymer (negative and positive ions) will be arranged in a crystalline phase leading to a change in material polarization, and the main factor here is the material’s ability to reorient crystallites. While for amorphous polymers that do not contain crystallites, piezoelectricity is related to the orientation of the molecular dipoles, in these polymers, the polarization is quasi-stable compared to semi-crystalline ones. The activation of piezoelectricity and molecular dipole orientation in the amorphous polymers occurs slightly higher than their glass transition temperature.

In the piezoelectric composites, piezoelectric inorganic particles or pillars will be dispersed in a polymer matrix, such as a ceramic/polymer composite [37]. In these piezoelectric composites, a combination of the advantages of each material can be obtained, such as high coupling factor and dielectric constant of piezoelectric particles or pillars; and the mechanical flexibility, stiffness coefficient, low acoustic impedance thermal, and chemical stability of polymer matrix. In the voided charged polymers, the piezoelectric properties are related to the internal gas voids inside the polymer matrix. Under an electric field effect, gas molecules will be ionized, which leads to the rearrangement of the opposite charges and to their assembly on either side of the voids, leading to the formation of internal dipoles. This means that piezoelectricity will be induced as a result of any void deformation. The gas type, its pressure, void density, and shape are considered the main piezoelectric factors here [37].

The ferroelectric polymers have some advantages, such as showing a good mechanical energy density as a result of their high elastic modulus and fast response. As for the disadvantages of these polymers, they also need a relatively high voltage, complicated production, especially concerning thin films, and low strain [38].

### 2.3. Electrostrictive Graft Elastomers

Another type of dielectric electroactive polymers is electrostrictive graft elastomers, which consist of flexible macromolecular polymer chain; and grafted crystalline pendant groups, which by applying an external electric field can change their alignment as a result of the polarization, Figure 10. The grafted crystallites on the flexible backbone polymer chain can be cross-linked with neighboring ones and can create polarized monomers, which include atoms with partial charges and cause the dipole effect. These dipole points under the electric field can rotate the whole polymer unit, which in its terms leads to the electrostrictive strain and polymer deformation [39].

Chlorotrifluoroethylene-vinylidene fluoride is considered the famous component of electrostrictive graft elastomers as the main chain and vinylidene fluoride-trifluoroethylene (VDF-TrFE) is used as a polar graft component [40]. The ratio of each component in the electrostrictive graft elastomers and the preparation conditions have the main effect on the obtained properties. For example, a higher obtained elongation at constant electric field strength can be achieved by increasing the ratio of the polar grafted groups or by increasing the material crystallinity by annealing it for a long time [41]. In reference [42], the authors investigated poly(vinylidene fluoride-trifluoroethylene-chlorotrifluoroethylene) as an active layer in the soft actuators. They showed that the preparation of aligned electrospun nanofiber mat based on poly(VDF-TrFE-CTFE) led to lower electrical power consumption under activation by AC electric field with fast viscoelastic relaxation. The authors illustrated that using electro-spinning technology leads to obtaining higher surface area-to-volume ratio, higher porosity, and lower density compared to the material obtained by extrusion technology.

This type of EAP, which is considered cheap, has strain values up to 5% with fast response; however, as all-dielectric polymers, it needs high driving voltage to be activated [31,43].

### 2.4. Liquid Crystal Elastomers (LC)

Liquid crystal elastomers consist of an elastomeric backbone and cross-linked side-chains, which are prepared based on crosslinking reaction of reactive-mesogenic monomers. The working principle of these polymers is based on the loss and restoration of the molecular groups’ order (their orientation) as a result of the phase transitions under applying external stimuli, such as light, heat, or an electric field. The main character of liquid crystals is that they can merge the long-range order of solid crystals with the mobility and flexibility of the fluid phase [44,45]. LC elastomers show spontaneous ferroelectricity [46]. In reference [47], the authors illustrated the mechanism of the conversation of the electrical energy into the mechanical for a ferroelectric liquid crystalline polymer, in which its mesogens are linked to a polysiloxane backbone via flexible alkyl spacers in a comb-like mode and the side-chains of the polymer were crosslinked together. By applying an electric field, the mesogenic side groups will be polarized and turn by θ° with a decrease in the smectic layer, and after removing the electric field, these groups will return to their initial position and thickness as can be seen in Figure 11.

Liquid crystal elastomers produce high stress and strain by increasing the temperature, and they need lower electrical voltage 100 times compared to dielectric and ferroelectric EAPs (in the range of 1.5–5 kV), but they show a slow response [31].

One of the interesting applications of LC elastomers is the selective detection of SARS-CoV-2 [48]. In this work, researchers used liquid crystal films of cationic surfactant dodecyltrimethylammonium bromide (DTAB), which was decorated with cationic surfactant and complementary 15-mer single-stranded deoxyribonucleic acid (ssDNA) probe, to detect femtomolar concentrations of single-stranded ribonucleic acid (ssRNA) of SARS-CoV-2. As can be seen in Figure 12, when the ssDNA probe was adsorbed, the flexible ssDNA probe chains were spreading at the surface of LC and interacting with the DTAB; therefore, its polarization changed leading to a reorientation of the LC from homoerotic to either tilted or planar one.

## 3. Ionic Electroactive Polymers

The main work principle for the ionic EAPs is the diffusion of the ions, so that they can be activated by the presence of two electrodes and the electrolyte; i.e., the change of the actuator size and shape depends here on the mobility and diffusion of ions in the ionic liquids electrolyte, Figure 13. By applying an electric field, anions and cations begin to diffuse towards anode and cathode respectively, which will lead to actuator swelling and bending [49].

In contrast to dielectric EAPs, ionic EAPs need a low voltage to be activated, and they are bistable (i.e., they have two stable equilibrium states). Disadvantages of the ionic EAPs are connected with their wetness that has to be maintained, and the occurrence of electrolysis above a certain voltage, which leads to irreversible material damage; slow response; low electromechanical coupling efficiency; a low actuation force as a result of their bending; and under a DC voltage activation, it is difficult to conserve their constant displacement [14,50].

Depending on the conduction mechanism of ionic EAPs, they can be divided into intrinsic ones, which have electric conductivity by electron mobility or conductive particles into polymer matrix; and extrinsic ones, which have ionic conductivity, Figure 14.

### 3.1. Ionic Polymeric Gels

Ionic gels in general consist of polymer network chains and electrolyte solutions. These polymers are activated due to the changes in the environment from an alkaline to an acid and vice versa, which causes the swelling or shrinkage of the gel. This type of chemical reaction is called a chemo-mechanical reaction [51]. The working principle of this type of EAP is that by applying a voltage, a movement of hydrogen ions in or out of the gel will occur like the reaction between an acid and an alkaline [31]. Polyvinyl alcohol and poly(2-acrylamide-2-methyl-1-propanesulfonic acid) are known polymers that form gels and are used as ionic EAPs. This type of EAP needs low driving voltage and it can match the mechanical and the energy density of biological muscles, but its responsiveness is considered slow [52]. This EAP type can be classified as aqueous (hydrogels) and non-aqueous ionic gels depending on the solvent types in the polymer network [53]. Hydrogels are formed by polymerization of vinyl monomers in electrolyte solutions by adding cross-linkers and initiators [54], while non-aqueous ones are prepared by swelling polymer networks using organic electrolyte solutions that exhibit physical interactions in polymer networks [55].

Since the main limitation of hydrogels is their weak mechanical properties, an interesting example of hydrogels is presented in reference [56], in which highly stretchable and tough hydrogels are based on alginate by using ionic and covalent crosslinks, Figure 15. As the authors explained, a high modulus can be achieved depending on covalent bonds that provide constantly strong crosslinks, while high energy dissipation can be insured by ionic bonds that act as reversible sacrificial bonds.

In reference [57], a composite sheet of oxidized multi-walled carbon nanotube (oxidized-MWNT)/polyvinyl alcohol (PVA) was prepared using a membrane filtration process. They presented that a composite of 30 wt.% PVA showed the best properties, which are a strain to failure of 5.5%, stress to failure of 51 MPa, Young’s modulus of 3.4 GPa, and conductivity of 9 S/cm. The authors explained that the porosity of their ionic gel films allows ions to migrate toward the nanotubes, which leads to volumetric changes in the gel volume. The sensor deformation is related to the swelling of the oxidized-MWNT/PVA gel sheet. This generated stress was obtained by applying voltages in the range of 2–10 V and can reach 1.8 MPa.

In the reference [58], a hydrogel of polyvinyl alcohol (PVA) and cellulose nanocrystal (CNC) was prepared using a freeze-thaw technique (Figure 16). Since the cellulose concentration is a significant factor to obtain a good dispersion in the PVA matrix, authors showed that by increasing the cellulose concentration the hydrogel compressive strength will be significantly decreased, which is caused by the insufficient cellulose dispersion. Moreover, CNC concentration also plays the main role in the actuation process. The authors showed that by increasing the CNC concentration, the gel displacement was improved, and they obtained the maximum strain under the electric field of 0.4 V/μm when the CNC concentration was 35 wt.%. The highest displacement of 14.4 μm was obtained at a voltage of 1.6 kV.

Nonaqueous ionic gels are prepared by using polymer gelation in ionic liquids with non-covalent associations such as hydrogen bonding, phase separation, and supra-molecular interaction. However, using a large amount of harmful organic solvents to prepare this type of ionic gel is considered the main disadvantage [53]. In reference [59], the authors prepared a non-aqueous liquid electrolyte based on 1-ethyl 3-methylimidazolium bis(nonafluorobutane-1-sulfonyl imidate) ionic liquid, Figure 17. The author showed that by improving electrical ion migration pathways, the obtained gel conductivity was improved up to 2.21 mScm^−1^ under a voltage of 4.3 V.

### 3.2. Conductive Polymers

Conductive polymers are materials which respond to an applied voltage by swelling/compression, Figure 18. This response is related to the oxidation or reduction of the material, which depends on its polarity. As can be seen in Figure 18, in the oxidation process, by removing electrons and insertion of anions, a behavior of crosslinking between the generated polarons and dopant anions will occur, which leads to a swell in the actuator. In the reduction process by applying a negative voltage, the compression/swelling response is dependent on the neutralized anion’s size and mobility; if the dopant anion is small and mobile (Figure 18a), anions and solvent molecules will be exiled out of the polymer to preserve its neutralization, which leads to a polymer shrinkage, while if the dopant anion is large and immobile (Figure 18b), anions cannot leave the polymer matrix during the reduction, so the cation in the solution will move toward the polymer to neutralize anions, which leads to a further expansion in the polymer matrix [60,61].

Hydrogels based on conductive polymers have some advantages, such as low voltage needed, high conductivity, good biological compatibility, high water content, and hierarchical interconnected structure [62,63,64]. Polyaniline (PANI) and polypyrrole (PPy) are known examples of conductive polymers, which, by using various crosslinking agents, such as phytic acid [65,66], sodium dodecyl benzene sulfonate [67], and amino trimethylene phosphonic acid [68,69], conductive polymer hydrogels can be prepared. On the other hand, this type of EAP shows slow response, fatigue after repeated activation, and deterioration under cyclic actuation [52].

In reference [70], researchers prepared conductive hydrogels using poly(3,4-ethylenedioxythiophene) polystyrene sulfonate (PEDOT:PSS) as a conductive dopant to crosslink the conductive polymers (polyaniline, polypyrrole (PPy), and poly-aminoindole), Figure 19. The electrostatic interaction between the positively charged conductive polymer chains and the negatively charged SO^3−^ groups in the conductive dopant was the basis of the preparation mechanism of these hydrogels. Authors showed that all prepared hydrogels have high electrical conductivity, and the highest value of 70.54 S/cm was obtained for the PPy/PEDOT:PSS hydrogel. Moreover, the authors showed that all prepared hydrogels have an elastic modulus in the range of 0.2–5.0 kPa, which corresponds to the mechanical properties of biological tissues. In addition, the authors illustrated that the use of their prepared hydrogels as electrochemical biosensors to detect dopamine and hydrogen peroxide showed a wide linear range, low sensing limits, and decent anti-interference properties.

In reference [71], researchers prepared composite pellets using carbon nanotubes (CNT) and poly(methyl methacrylate) (PMMA) by using the electrostatic assembly method to obtain homogeneous decoration of CNT onto PMMA particles, Figure 20. Using a small amount of CNT, the obtained composites showed an enhanced electrical conductivity in the range of 10^−5^–10^−1^ S/m with CNT contents of 0.0068 to 0.045 vol.%, respectively, compared to the pure PMMA with an electrical conductivity of 10^−15^ S/m.

### 3.3. Ionic Polymer–Metal Composites

In general, ionic polymer–metal composites consist of a polymeric ion exchange membrane with negative ions fixed on interconnected clusters, fully swollen by water, and a thin layer of noble metal as electrodes, Figure 21. By applying a voltage on this EAP, the hydrated cation begins to move freely towards the cathode through the polymer membrane, which provides channels for their movements; and then cation accumulates on the cathode, causing swelling of the actuator and its bending, Figure 21 [52,72]. The needed low driving voltage (1–5 V), the high ionic conductivity of the electrolyte, fast response, and large deformation are considered the main advantages of this EAP type. On the other hand, electrolysis of electrolytes based on water causing a reduction in ion mobility, cracking of electrodes causing low cycle life, and high dehydration are considered the main disadvantages of ionic polymer–metal composites [73,74]. Since gold, platinum, and palladium exhibit excellent electrical conductivity and high electrochemical stability, they are usually used as electrodes [75,76]. Moreover, Nafion (sulfonated tetrafluoroethylene based fluoropolymer-copolymer) is considered one of the main polymer bases of ionic polymer–metal composites.

In the water-based ionic polymer–metal composites, when an electric field is applied between metallic electrodes, molecules of water connected with positive ions begin to rapidly migrate toward the cathode, which leads to a deformation in the actuator and its bending due to the distribution of the water molecules. Under full saturation, this anode deformation is followed by a slow back-relaxation. This means that the number of water molecules plays a significant role in this deformation-relaxation process [77]. The main problem of the water-based ionic polymer–metal actuator, which reduces both actuator efficiency and its lifetime, is the leakage of the water molecules into the cracks of the metal electrodes as a result of the electrolysis and the evaporation of water in the presence of air [77]. In the reference [72], the authors tried to solve this problem by adding polyethylene oxide (PEO) into a Nafion matrix, which has a significant capacity for water retention. They showed that using PEO by 20 wt.% led to a reinforcement of the electromechanical performances and a restriction of the displacement attenuation of the actuator. Authors presented that the addition of PEO led to enhancing the peak-to-peak displacement of the PEO/Nafion actuator two times compared to pure Nafion one in the conditions of 2 V of DC voltage within 25 s. Moreover, the authors showed a good enhancement in the volumetric work density and an increase in the working time in the air.

## 4. Conclusions and Prospects

In this article, the latest studies on electroactive polymer-based actuators and their work principles were reviewed. The applications of these materials as biomimetic robotics, sensors, and actuators in medical applications, such as artificial muscles, drug delivery, and antimicrobial materials, are considered trending topics in this period. To apply EAPs in biomedical applications, their interesting properties should be optimized, such as mechanical properties, conductivity, porosity, viscosity, hydrophobicity, and biodegradability. The main properties of most EAP types such as driving voltage, electrically induced stress, strain, response speed, elastic modulus, and power density are presented in Table 1. Moreover, the main advantages and disadvantages of the main EAP types are presented in Table 2.

Understanding the EAP work principles is considered the main factor for applying them, which means optimizing the polymer architecture with required structural modification and optimization of the EAP processing and programming technology. The main restrictions of these materials to be applied in biomedical applications are the activation method of actuators and the realization of the required mechanical properties for a specific application with a long lifetime and rapid response of cycle activation to mimic human ones.

As can be seen in Table 1 and Table 2, for dielectric EAPs, decreasing the driving voltage and enhancing their production methods to prepare thin films are considered the main issues that need to be solved in future investigations. On the other hand, improving the ionic EAPs mechanical properties and accelerating their actuation response will be the essential directions for the investigations on further development. Moreover, enhancement of ion mobility in extrinsic EAPs by structure modification of polymers or electrolytes, and enhancement of electric conductivity by electron mobility or conductive particles in intrinsic EAPs can be taken into account as attractive topics for the researchers.

In conclusion, in spite of several decades of R&D EAPs, to employ them in medical applications is currently far from a reality, such as their performance and response, long-term stability, and mechanical properties that still need further development to achieve EAPs actuators with properties that mimic the requirements of each needed application. In addition, the preparation methods of EAP actuators are also considered a main challenge that needs further investigation to improve the characteristics of EAP actuators for use in real applications. For example, 3D printing techniques for EAPs with accurate dimensional control can lead to a revolution in EAP developments in order to apply them in future soft robots, artificial muscles, and tissue engineering applications [78,79].

## Figures and Tables

**Figure 1 nanomaterials-12-02272-f001:**
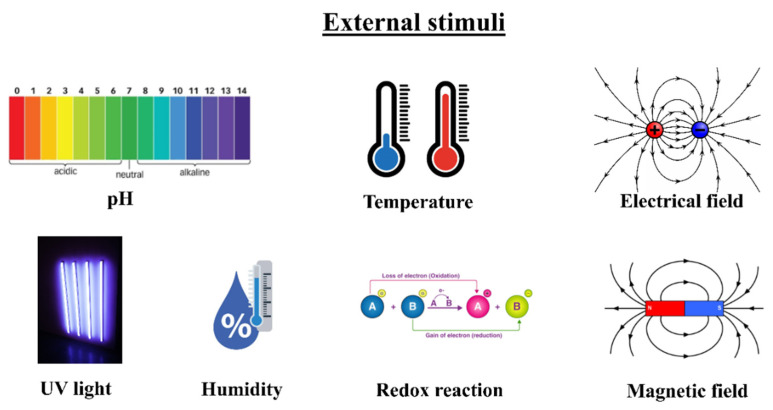
External stimuli for activation of polymer actuators.

**Figure 2 nanomaterials-12-02272-f002:**
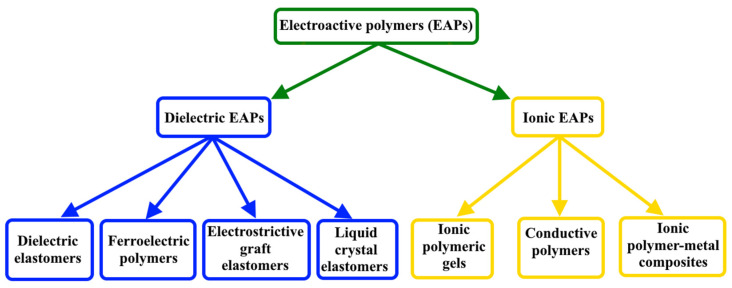
Electroactive polymers groups.

**Figure 3 nanomaterials-12-02272-f003:**
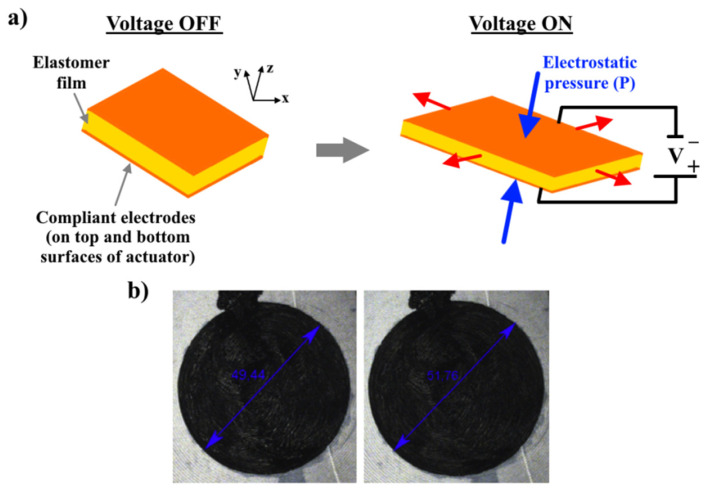
(**a**) Work principle of the DE actuator and (**b**) actuation of 3D printed DE actuator (U = 4.67 kV). Reprinted/adapted with permission from Ref. [16]. 2019, Elsevier.

**Figure 4 nanomaterials-12-02272-f004:**
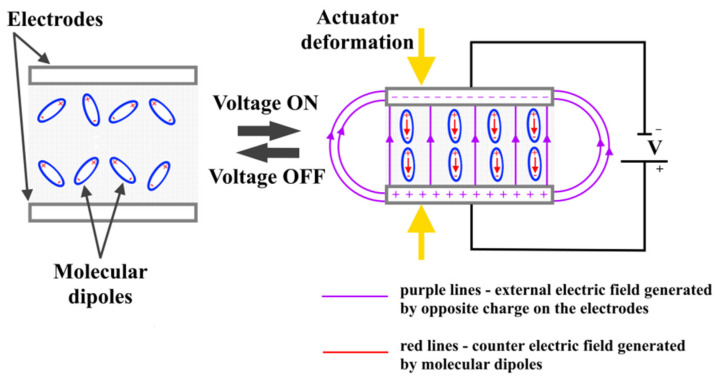
Polarization process of molecular dipoles of elastomer films under an external electric field.

**Figure 5 nanomaterials-12-02272-f005:**
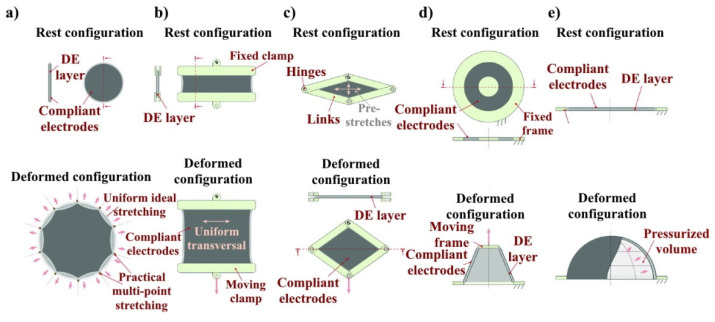
Some topologies of DEG. (**a**) Equibiaxial DEG (ideal uniformly stretched generator, and practical multipoint stretching embodiment); (**b**) pure shear (namely, strip-biaxial) DEG; (**c**) diamond DEG; (**d**) cone DEG; and (**e**) circular diaphragm DEG [21].

**Figure 6 nanomaterials-12-02272-f006:**
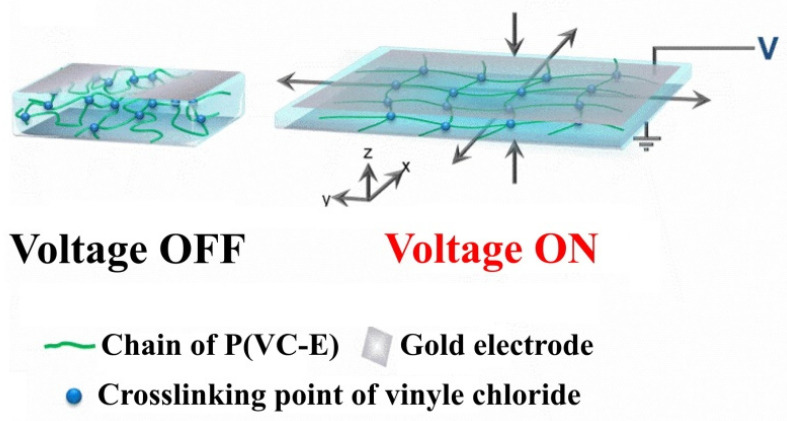
Schematic of P(VC-E)-4 deformation mechanism driven by the high electric field. Reprinted/adapted with permission from Ref. [27]. 2022, Elsevier.

**Figure 7 nanomaterials-12-02272-f007:**
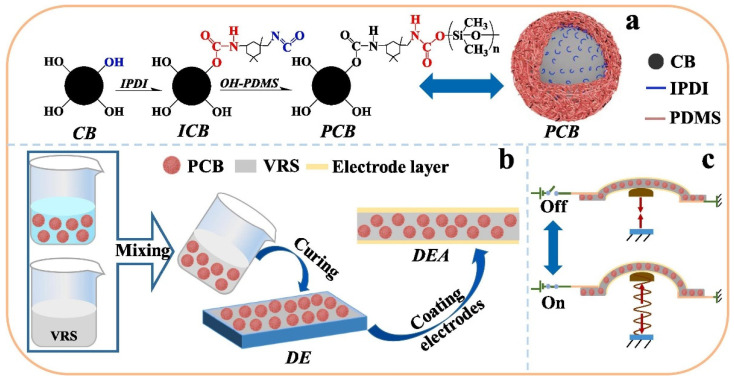
(**a**) Illustration of the formation of PCB particles. Catalyzed by DBTDL, hydroxyl group of CB surface initially crosslinks with IPDI to produce a N=C=O group functionalized CB, then couples with OH-PDMS, thus coating a layer of PDMS film to CB surface. (**b**) Illustration of the fabrication of DEA. The PCB particles are filled in VRS precursor and cured to produce a PCB/VRS hybrid film. The conductive silicone paste is brushed on both PCB/VRS film surfaces to make a trilayer DEA. (**c**) Prestrained by an elastic spring, the hybrid DEA takes an out-of-plane actuation under a pulse electrical field. Reprinted/adapted with permission from Ref. [28]. 2022, Elsevier.

**Figure 8 nanomaterials-12-02272-f008:**
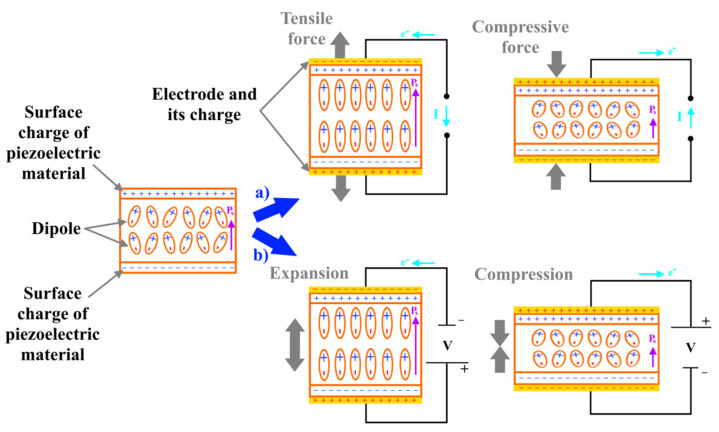
Schematic representation of piezoelectric effects depending on stress and voltage: (**a**) applying a stress, (**b**) applying a voltage.

**Figure 9 nanomaterials-12-02272-f009:**
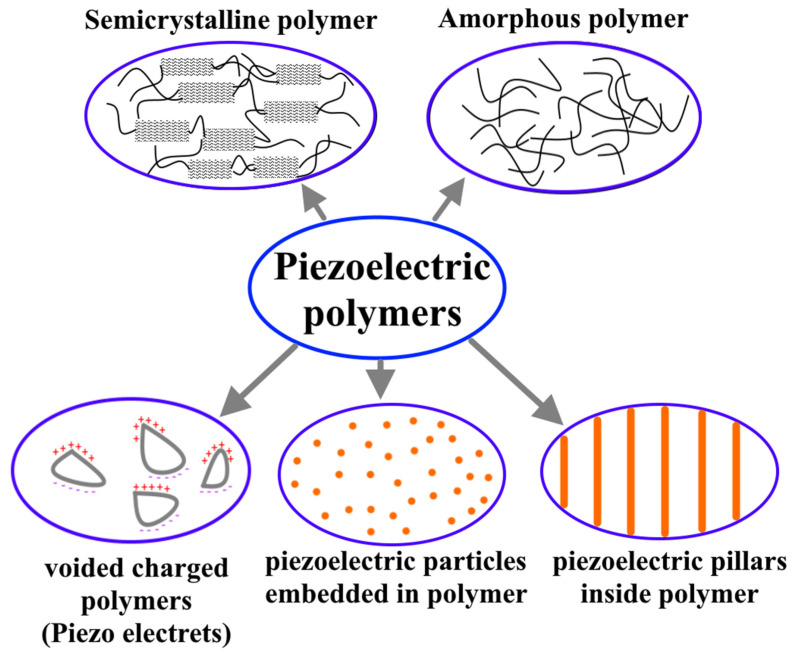
Piezoelectric polymers classification.

**Figure 10 nanomaterials-12-02272-f010:**
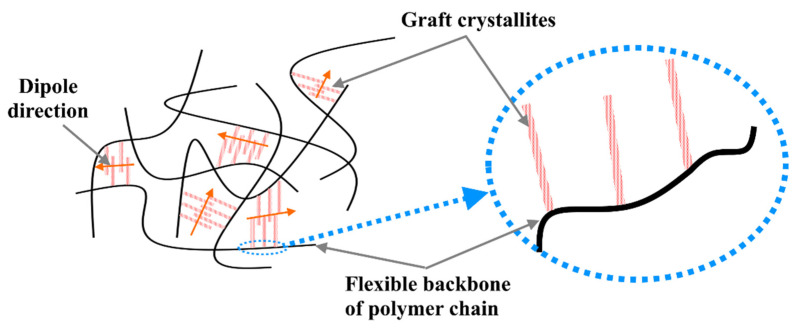
Schematic structure of electrostrictive graft elastomers.

**Figure 11 nanomaterials-12-02272-f011:**
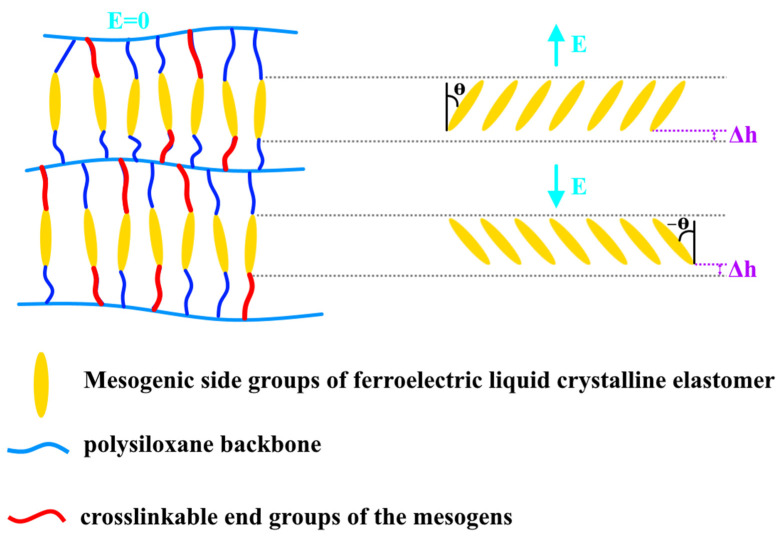
The electroclinic effect in ferroelectric liquid crystalline elastomers. The polarized mesogenic side groups will turn by “θ” or “−θ” depending on the direction of the electric current flow E (electric field out of or into the plane of material). Δh—a decrease in the thickness of the polarized mesogenic side groups during the application of the electric field.

**Figure 12 nanomaterials-12-02272-f012:**
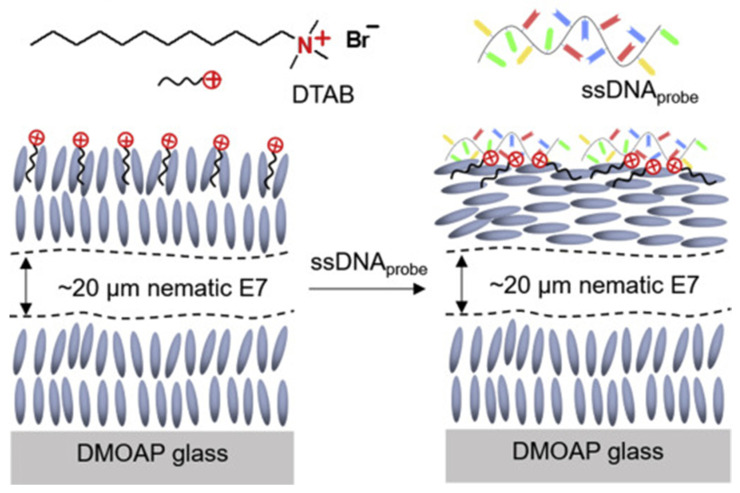
Schematic illustration of the optical response of the DTAB-decorated LC film to the adsorption of the ssDNA probe. Reprinted/adapted with permission from Ref. [48]. 2022, Elsevier.

**Figure 13 nanomaterials-12-02272-f013:**
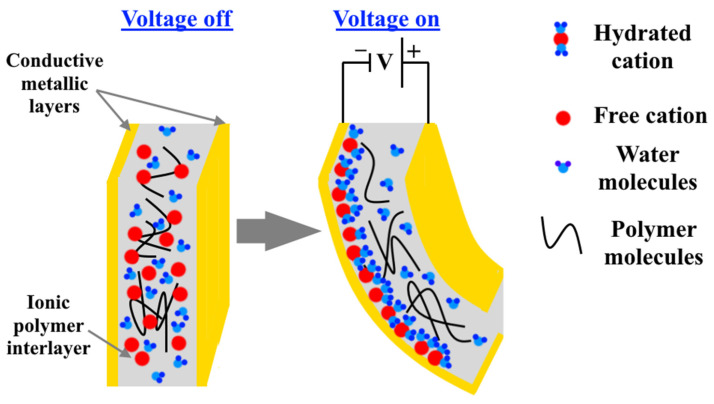
Work principle for the ionic EAPs.

**Figure 14 nanomaterials-12-02272-f014:**
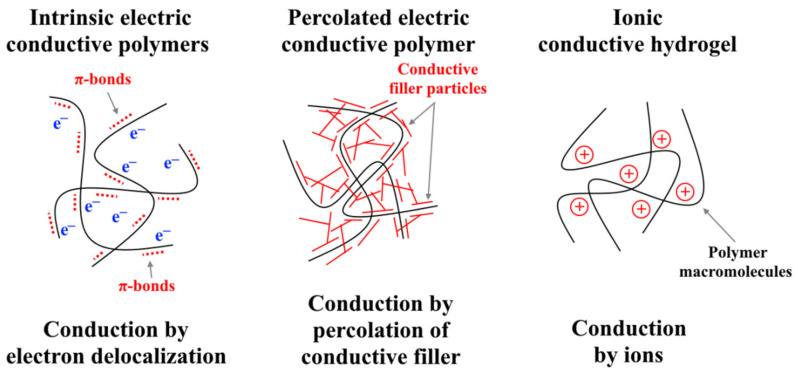
Conduction mechanisms of electroactive polymers.

**Figure 15 nanomaterials-12-02272-f015:**
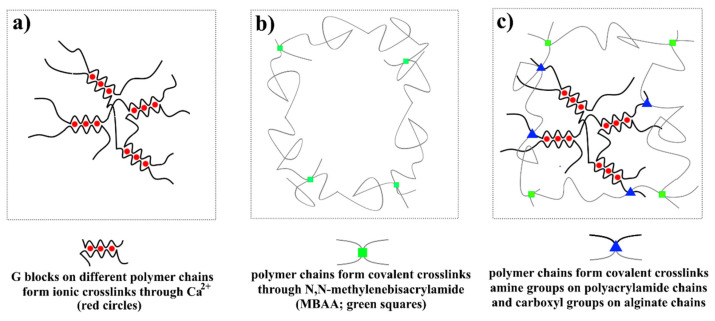
Schematics of three types of hydrogels based on alginate: (**a**) ionically crosslinked alginate chains, (**b**) covalently crosslinked polyacrylamide chains, (**c**) covalently crosslinked amine groups on polyacrylamide chains and carboxyl groups on alginate chains.

**Figure 16 nanomaterials-12-02272-f016:**
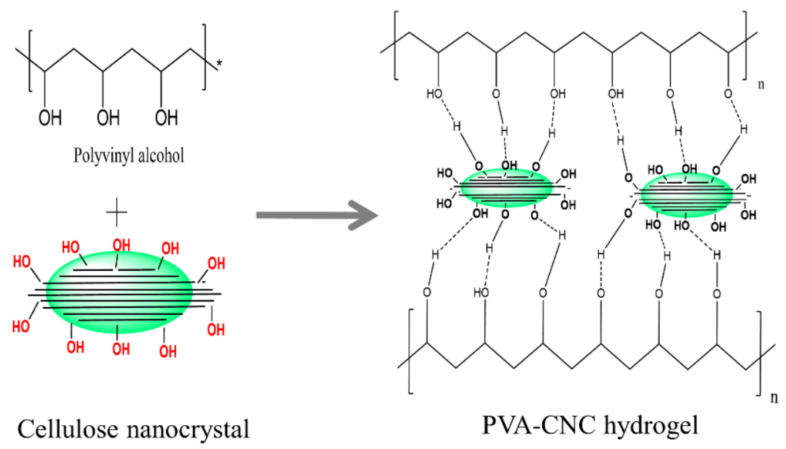
Formation of the PVA-CNC hydrogel [58].

**Figure 17 nanomaterials-12-02272-f017:**
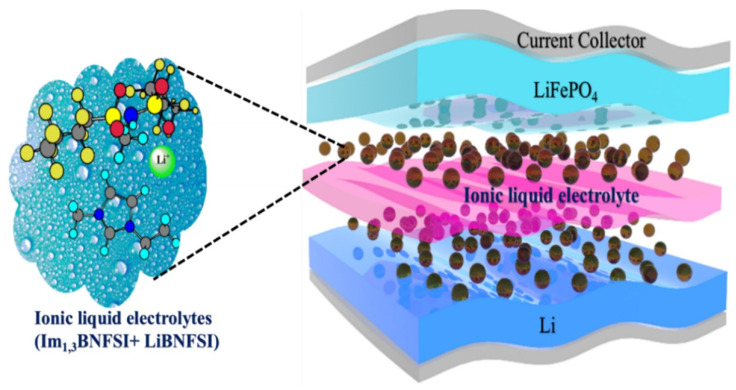
Non-aqueous liquid electrolytes based on 1-ethyl 3-methylimidazolium bis(nonafluorobutane-1-sulfonyl imidate) ionic liquid. Reprinted/adapted with permission from Ref. [59]. 2020, Elsevier.

**Figure 18 nanomaterials-12-02272-f018:**
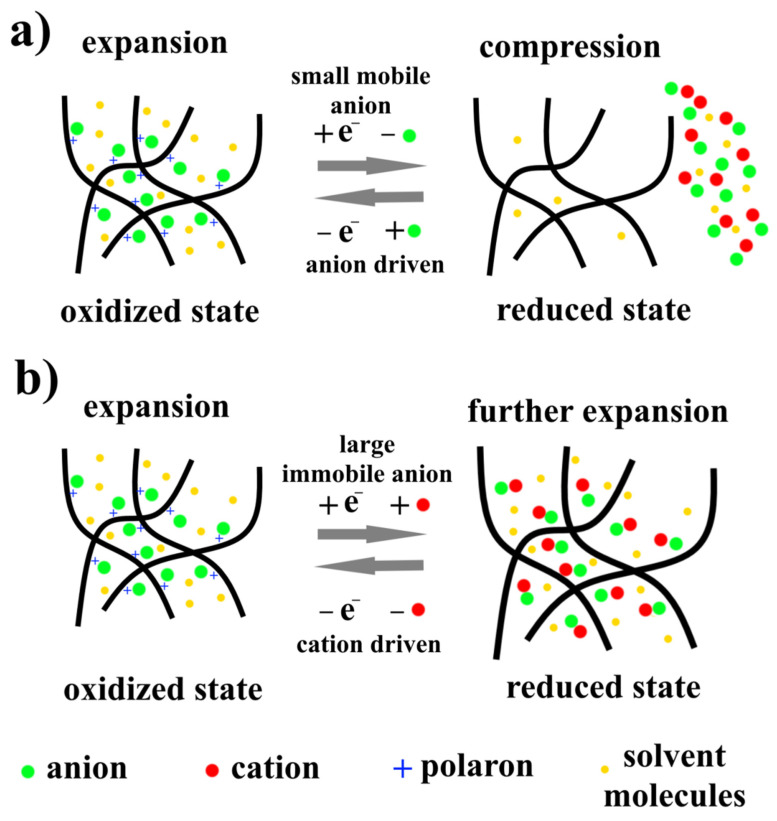
Ionic actuation mechanisms of conductive polymer actuators depending on the anion size and mobility: (**a**) small anion mobile, (**b**) large and immobile anion.

**Figure 19 nanomaterials-12-02272-f019:**
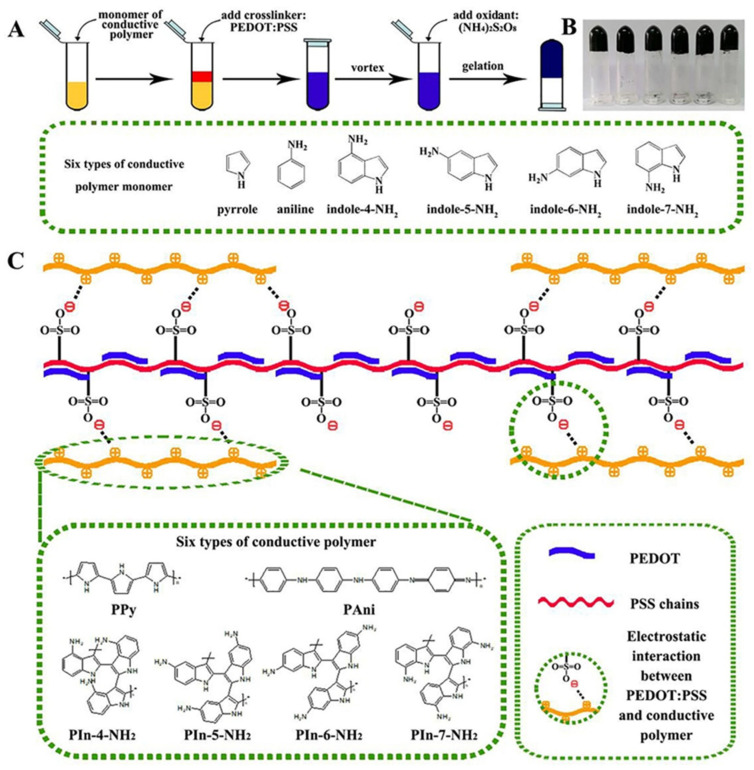
(**A**) Schematic illustration of gelation processes for six types of conductive polymer hydrogels: PPy, PAni, PIn-4-NH2, PIn-5-NH2, PIn-6-NH2, and PIn-7-NH2. (**B**) Photographs of the conductive polymer hydrogels (from left to right: PPy/PEDOT:PSS hydrogel, PAni/PEDOT:PSS hydrogel, PIn-4-NH2/PEDOT:PSS hydrogel, PIn-5-NH2/PEDOT:PSS hydrogel, PIn-6-NH2/PEDOT:PSS hydrogel, PIn-7-NH2/PEDOT:PSS hydrogel). (**C**) Crosslinking mechanism of the conductive polymer/PEDOT:PSS hydrogels. The addition of the PEDOT:PSS as the dopant and gelator possess a large amount of negatively charged sulfonic acid functional group, which can form electrostatic interaction with the positively charged conductive polymer chains. Reprinted/adapted with permission from Ref. [70]. 2020, Elsevier.

**Figure 20 nanomaterials-12-02272-f020:**
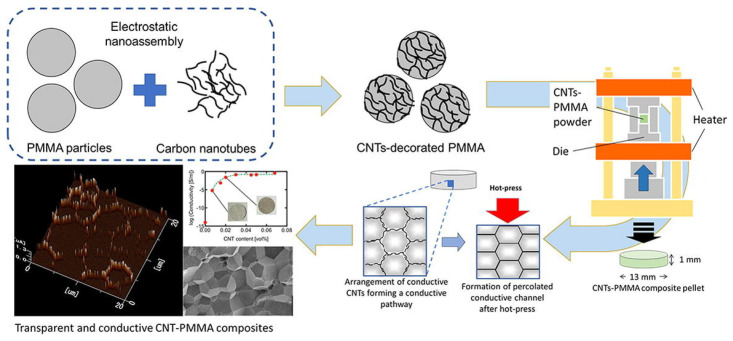
Electrostatically nano-assembled carbon-nanotubes-decorated poly(methyl methacrylate) (PMMA) particles for fabrication of transparent conductive polymer composites. Reprinted/adapted with permission from Ref. [71]. 2020, Elsevier.

**Figure 21 nanomaterials-12-02272-f021:**
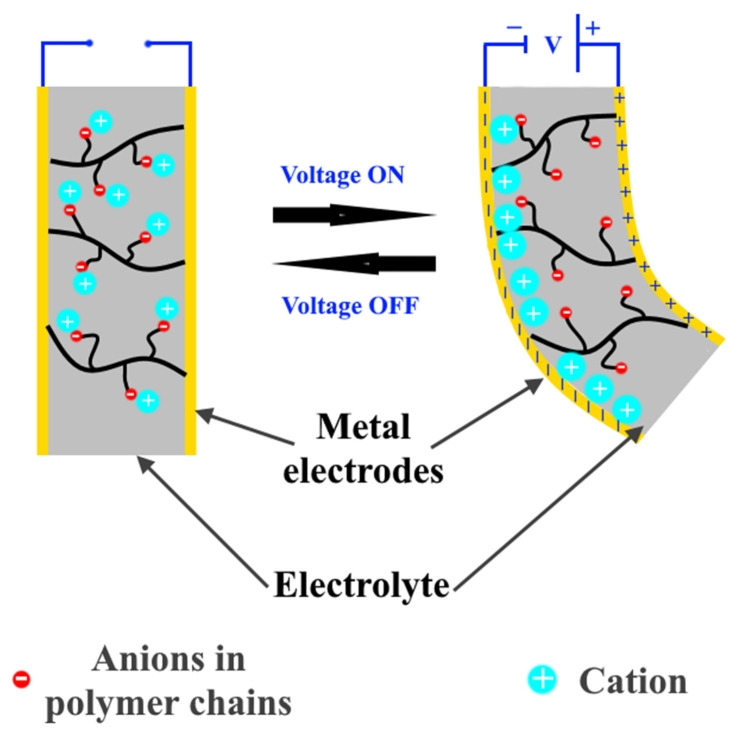
Work principle of ionic polymer–metal composites.

**Table 1 nanomaterials-12-02272-t001:** Characteristics of electroactive polymers (EAPs).

Characteristic	Dielectric Elastomers [15,21,26,27,28,29,30]	Piezoelectric Polymers [10,15,36,37,38]	Conductive Polymers [15,60,61,62,63,64,65,66,67,68,69,70,71]	Polymer-Metal Composites [72,73,74,75,76,77]	Hydrogels [15,51,52,53,54,55,56,57,58]
Driving voltage (V)	300–10,000	50–20,000	1–30	1–10	1–30
Typical strain, %	25–400	0.1–10	1000	0.5–30	<10
Typical stress (MPa)	3–38	Up to 20	Up to 200	3–78	30–55
Typical specific elastic energy density (J/g)	0.1–3.4	0.0013–30	0.1–1	Up to 0.004	Up to 0.01
Efficiency (%)	Up to 80	Up to 90	Up to 95%	Up to 5	Up to 80%
Relative speed (full cycle)	Fast (up to 1000 Hz)	Fast (up to 200)	Slow (up to 15)	Slow (<40 Hz)	Slow (around 1 Hz)

**Table 2 nanomaterials-12-02272-t002:** Advantages and disadvantages of electroactive polymers (EAPs).

Material	Advantages	Disadvantages
Dielectric elastomers	large displacements of strain high energy density fast response (up to 1000 Hz) cheap	high driving voltage the occurrence of pre-strain and compromise in the actuation force due to their high displacements
Piezoelectric polymers	good mechanical energy density high elastic modulus fast response	require a relatively high voltage; complicated production, especially as a thin film low strain
Liquid Crystal Elastomers (LC)	high stress high strain require lower electrical voltage 100 times compared to dielectric and ferroelectric EAPs (in the range of 1.5–5 kV)	slow response hysteresis
Ionic Hydrogels	can mimic the strength and energy density of biological muscles require low voltage bistable	weak mechanical properties slow response their wetness has to be maintained the occurrence of electrolysis above a certain voltage low response low electromechanical coupling efficiency
Polymer–metal composites	require low driving voltage high ionic conductivity of the electrolyte fast response large deformation	electrolysis of electrolytes based on water causing a reduction in ion mobility cracking of electrodes causing low cycle life high dehydration
Conductive polymers	require low voltage high conductivity good biologically compatible high water content hierarchical interconnected structure	slow response fatigue after repeated activation deterioration under cyclic actuation

## Data Availability

No additional data available.

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
