# Peer review of "Electroactive Polymer-Based Composites for Artificial Muscle-like Actuators: A Review"

_nanomaterials, 2022, doi:10.3390/nano12132272_

Round 1
Reviewer 1 Report
It can be publsihed with following revision since this paper generally looks well-organized
1. Abstract should have modern day values from electroactive-polymers
2. Figures 5 and 6 should be replaced with better visible picture
3. I could not find mechanical stress on the electroactive materials which should be placed in a separate para.
4. Figure 17 is inconsistent with font if you compare with other figures
5. Figure 19 has similar issues with Figure 17
6. Paper should have outlook and perspective section that would end with some special proposal for future readers.
Reference should be updated without doubts
Author Response
- Abstract should have modern day values from electroactive-polymers
Thank you for your kind comment. Please, the following abstract:
“Unlike traditional actuators, such as piezoelectric ceramic or metallic actuators, polymer actuators are currently attracting more interest in biomedicine due to their unique properties, such as light weight, easy processing, biodegradability, fast response, large active strains, and good mechanical properties. They can be actuated under external stimuli, such as chemical (pH changes), electric, humidity, light, temperature, and magnetic field. Electroactive polymers (EAPs), called ‘artificial muscles’, can be activated by an electric stimulus, and fixed into a temporary shape. Restoring their permanent shape after the release of an electrical field, electroactive polymer is considered the most attractive actuator type because of its high suitability for prosthetics and soft robotics applications. However, robust control, modeling non-linear behavior and scalable fabrication are considered the most critical challenges for applying the soft robotic systems in real conditions. Researchers, from around the world, investigate the scientific and engineering foundations of polymer actuators, especially the principles of their work, for the purpose of a better control of their capability and durability. The activation method of actuators and the realization of required mechanical properties are the main restrictions on using actuators in real applications. The latest highlights, operating principles, perspectives, and challenges of electroactive materials (EAPs) such as dielectric EAPs, ferroelectric polymers, electrostrictive graft elastomers, liquid crystal elastomers, ionic gels, ionic polymer-metal composites are reviewed in this article.”
- Figures 5 and 6 should be replaced with better visible picture
Thank you for your kind comment. Figures 5 and 6 were enhanced as follows:
- I could not find mechanical stress on the electroactive materials which should be placed in a separate para.
The main properties of the most of EAPs types such as driving voltage, electrically induced stress, strain, response speed, elastic modulus and power density are presented in Table 1
Table 1. Characteristics of electroactive polymers (EAPs)
|
Characteristic |
Dielectric elastomers [15, 21, 26-30] |
Piezoelectric polymers [15, 36-38, 78] |
Conductive polymers [15, 60-71] |
Polymer-metal composites [72-77] |
Hydrogels [15, 51-58] |
|
Driving voltage (V) |
300-2000 |
50-20000 |
1-30 |
1-10 |
1-30 |
|
Typical strain, % |
25-400 |
0.1-10 |
1000 |
0.5-30 |
<10 |
|
Typical stress [MPa] |
3-38 |
Up to 20 |
Up to 200 |
3-78 |
30-55 |
|
Typical specific elastic energy density [J/g] |
0.1-3.4 |
0.0013-30 |
0.1-1 |
Up to 0.004 |
Uo to 0.01 |
|
Efficiency [%] |
Up to 80 |
Up to 90 |
Up to 95% |
Up to 5 |
Up to 80% |
|
Relative speed (full cycle) |
Fast (up to 1000 Hz) |
Fast (up to 200) |
Slow (up to 15) |
Slow (<40 Hz) |
Slow (around 1 Hz) |
- Figure 17 is inconsistent with font if you compare with other figures
Figure 17 was taken from the reference [59] with a copyright permission without any editing.
- Figure 19 has similar issues with Figure 17
In Figure 17, a non-aqueous liquid electrolytes based on 1-ethyl 3-methylimidazolium bis(nonafluorobutane-1-sulfonyl imidate) ionic liquid was presented as an example for non-aqueous ionic gels and the work principle here is based on electrical ion migration; while Figure 19 presented gelation processes for six types of conductive polymer hydrogels using DI water and PEDOT:PSS aqueous solution for obtaining hydrogels. In addition, the work principle for both non-aqueous and aqueous gels is based on electrical ion migration.
- Paper should have outlook and perspective section that would end with some special proposal for future readers.
Thank you for your kind comment. Please, check the reformulated “Conclusion and prospects” section:
“In this article, the latest researches on electroactive polymer-based actuators and their work principles were reviewed. The applications of these materials as biomimetic robotics, sensors, and actuators in medical applications, such as artificial muscles, drug delivery, and antimicrobial materials, are considered trending topics in this period. To apply EAPs in biomedical applications, their interesting properties should be optimized, such as mechanical properties, conductivity, porosity, viscosity, hydrophobicity and biodegradability. The main properties of the most of EAPs types such as driving voltage, electrically induced stress, strain, response speed, elastic modulus and power density are presented in Table 1. Moreover, the main advantages and disadvantages of the main EAPs types are presented in Table 2.
Table 1. Characteristics of electroactive polymers (EAPs)
|
Characteristic |
Dielectric elastomers [15, 21, 26-30] |
Piezoelectric polymers [15, 36-38, 78] |
Conductive polymers [15, 60-71] |
Polymer-metal composites [72-77] |
Hydrogels [15, 51-58] |
|
Driving voltage (V) |
300-10000 |
50-20000 |
1-30 |
1-10 |
1-30 |
|
Typical strain, % |
25-400 |
0.1-10 |
1000 |
0.5-30 |
<10 |
|
Typical stress [MPa] |
3-38 |
Up to 20 |
Up to 200 |
3-78 |
30-55 |
|
Typical specific elastic energy density [J/g] |
0.1-3.4 |
0.0013-30 |
0.1-1 |
Up to 0.004 |
Uo to 0.01 |
|
Efficiency [%] |
Up to 80 |
Up to 90 |
Up to 95% |
Up to 5 |
Up to 80% |
|
Relative speed (full cycle) |
Fast (up to 1000 Hz) |
Fast (up to 200) |
Slow (up to 15) |
Slow (<40 Hz) |
Slow (around 1 Hz) |
Understanding the EAPs' work principles is considered the main factor for applying them, which means optimizing the polymer architecture with required structural modification and optimization of the EAPs processing and programming technology. The main restrictions of these materials to be applied in biomedical applications are the activation method of actuators and the realization of the required mechanical properties for a specific application with a long lifetime and rapid response of cycle activation to mimic human ones.
As can be seen in Tables 1 and 2, for dielectric EAPs, decreasing the driving voltage and enhancing their production methods to prepare thin films are considered the main issues that need to be solved in future investigations. On the other hand, improving the ionic EAPs mechanical properties and accelerating their actuation response will be the essential directions for the investigations on further development. Moreover, enhancement of ion mobility in extrinsic EAPs by structure modification of polymers or electrolytes, and enhancement of electric conductivity by electron mobility or conductive particles in intrinsic EAPs can be taken into account as attractive topics for the researchers.
Table 2. Advantages and disadvantages of electroactive polymers (EAPs)
|
Material |
Advantages |
Disadvantages |
|
Dielectric elastomers |
· large displacements of strain; · high energy density; · fast response (up to 1000 Hz); · cheap; |
· high driving voltage; · the occurrence of pre-strain and compromise in the actuation force due to their high displacements; |
|
Piezoelectric polymers |
· good mechanical energy density; · high elastic modulus; · fast response; |
· require a relatively high voltage; · complicated production, especially as a thin film; · low strain; |
|
Liquid Crystal Elastomers (LC) |
· high stress; · high strain; · require lower electrical voltage 100 times compared to dielectric and ferroelectric EAPs (in the range of 1.5-5 kV); |
· slow response; · hysteresis; |
|
Ionic Hydrogels |
· can mimic the strength and energy density of biological muscles; · require low voltage; · bistable; |
· weak mechanical properties; · slow response; · their wetness has to be maintained; · the occurrence of electrolysis above a certain voltage; · low response; · low electromechanical coupling efficiency; |
|
Polymer-metal composites |
· require low driving voltage; · high ionic conductivity of the electrolyte; · fast response; · large deformation; |
· electrolysis of electrolytes based on water causing a reduction in ion mobility; · cracking of electrodes causing low cycle life; · high dehydration; |
|
Conductive polymers |
· require low voltage; · high conductivity; · good biologically compatible; · high water content; hierarchical interconnected structure; |
· slow response; · fatigue after repeated activation; deterioration under cyclic actuation; |
In conclusion, up till now, in spite of several decades of R&D EAPs, to employ them in medical applications is far from real now, such as their performance and response, long-term stability and mechanical properties that still need further development to achieve EAPs actuators with properties that mimic the requirements of each needed application. In addition, the preparation methods of EAPs actuators are also considered a main challenge that need further investigation to improve the characteristics of EAP actuators for using them in real applications. For example, 3D printing technique for EAPs with accurate dimensional control can lead to a revolution in EAPs developments in order to apply them in future soft robots, artificial muscles, and tissue engineering applications [79, 80].”
Reference should be updated without doubts.
Thank you for your comment. We have added some new references to the manuscript. Kindly be informed that most of the used references are published from 2020 to the present.

Reviewer 2 Report
Artificial muscles-Like Actuators are the most important and the main application of polymeric actuators. In this review, the authors illustrated the actuatable materials can respond as a controllable change in their size or shape under an impact of external stimuli, such as chemical stimulus (pH changes), electric, humidity, light, temperature, magnetic field, etc.. The paper is very interesting and has great guiding significance for the development of related actuators. However, there are still many problems in the paper, which need to be revised.
1. Many pictures are not clear enough, for example: Figures 5, 6, 20.
2. The authors only list a lot of work at this stage, and it is recommended that the authors compare the advantages and disadvantages of different methods in detail.
3. The authors should quantitatively compare and analyze some important parameters such as deformation size and mechanical properties.
4. It is recommended to express the advantages and disadvantages of various methods intuitively through tables or graphs.
5. The authors should summarize more experience and make appropriate outlook based on the previous description and analysis in the conclusion part.
Author Response
- Many pictures are not clear enough, for example: Figures 5, 6, 20.
Thank you for your kind comment. Please, be informed that all figures resolutions were enhanced.
- The authors only list a lot of work at this stage, and it is recommended that the authors compare the advantages and disadvantages of different methods in detail.
Thank you for your kind comment. Please, check the added Tables 1 and 2 in the “Conclusion and prospects” section, where you can find a comparison between the advantages and disadvantages of different actuator types based on EAPs.
- The authors should quantitatively compare and analyze some important parameters such as deformation size and mechanical properties.
Thank you for your kind comment. Please, be informed that this information was added in Tables 1.
- It is recommended to express the advantages and disadvantages of various methods intuitively through tables or graphs.
Thank you for your kind comment. Unfortunately, to discuss the preparation method in details for each type of EAP actuators needs a separate work. Kindly, be informed that we plan to investigate and update this information in our next review.
- The authors should summarize more experience and make appropriate outlook based on the previous description and analysis in the conclusion part.
Thank you for your kind comment. Please, check the reformulated conclusion as follows:
“In this article, the latest researches on electroactive polymer-based actuators and their work principles were reviewed. The applications of these materials as biomimetic robotics, sensors, and actuators in medical applications, such as artificial muscles, drug delivery, and antimicrobial materials, are considered trending topics in this period. To apply EAPs in biomedical applications, their interesting properties should be optimized, such as mechanical properties, conductivity, porosity, viscosity, hydrophobicity and biodegradability. The main properties of the most of EAPs types such as driving voltage, electrically induced stress, strain, response speed, elastic modulus and power density are presented in Table 1. Moreover, the main advantages and disadvantages of the main EAPs types are presented in Table 2.
Table 1. Characteristics of electroactive polymers (EAPs)
|
Characteristic |
Dielectric elastomers [15, 21, 26-30] |
Piezoelectric polymers [15, 36-38, 78] |
Conductive polymers [15, 60-71] |
Polymer-metal composites [72-77] |
Hydrogels [15, 51-58] |
|
Driving voltage (V) |
300-10000 |
50-20000 |
1-30 |
1-10 |
1-30 |
|
Typical strain, % |
25-400 |
0.1-10 |
1000 |
0.5-30 |
<10 |
|
Typical stress [MPa] |
3-38 |
Up to 20 |
Up to 200 |
3-78 |
30-55 |
|
Typical specific elastic energy density [J/g] |
0.1-3.4 |
0.0013-30 |
0.1-1 |
Up to 0.004 |
Uo to 0.01 |
|
Efficiency [%] |
Up to 80 |
Up to 90 |
Up to 95% |
Up to 5 |
Up to 80% |
|
Relative speed (full cycle) |
Fast (up to 1000 Hz) |
Fast (up to 200) |
Slow (up to 15) |
Slow (<40 Hz) |
Slow (around 1 Hz) |
Understanding the EAPs' work principles is considered the main factor for applying them, which means optimizing the polymer architecture with required structural modification and optimization of the EAPs processing and programming technology. The main restrictions of these materials to be applied in biomedical applications are the activation method of actuators and the realization of the required mechanical properties for a specific application with a long lifetime and rapid response of cycle activation to mimic human ones.
As can be seen in Tables 1 and 2, for dielectric EAPs, decreasing the driving voltage and enhancing their production methods to prepare thin films are considered the main issues that need to be solved in future investigations. On the other hand, improving the ionic EAPs mechanical properties and accelerating their actuation response will be the essential directions for the investigations on further development. Moreover, enhancement of ion mobility in extrinsic EAPs by structure modification of polymers or electrolytes, and enhancement of electric conductivity by electron mobility or conductive particles in intrinsic EAPs can be taken into account as attractive topics for the researchers.
Table 2. Advantages and disadvantages of electroactive polymers (EAPs)
|
Material |
Advantages |
Disadvantages |
|
Dielectric elastomers |
· large displacements of strain; · high energy density; · fast response (up to 1000 Hz); · cheap; |
· high driving voltage; · the occurrence of pre-strain and compromise in the actuation force due to their high displacements; |
|
Piezoelectric polymers |
· good mechanical energy density; · high elastic modulus; · fast response; |
· require a relatively high voltage; · complicated production, especially as a thin film; · low strain; |
|
Liquid Crystal Elastomers (LC) |
· high stress; · high strain; · require lower electrical voltage 100 times compared to dielectric and ferroelectric EAPs (in the range of 1.5-5 kV); |
· slow response; · hysteresis; |
|
Ionic Hydrogels |
· can mimic the strength and energy density of biological muscles; · require low voltage; · bistable; |
· weak mechanical properties; · slow response; · their wetness has to be maintained; · the occurrence of electrolysis above a certain voltage; · low response; · low electromechanical coupling efficiency; |
|
Polymer-metal composites |
· require low driving voltage; · high ionic conductivity of the electrolyte; · fast response; · large deformation; |
· electrolysis of electrolytes based on water causing a reduction in ion mobility; · cracking of electrodes causing low cycle life; · high dehydration; |
|
Conductive polymers |
· require low voltage; · high conductivity; · good biologically compatible; · high water content; hierarchical interconnected structure; |
· slow response; · fatigue after repeated activation; deterioration under cyclic actuation; |
In conclusion, up till now, in spite of several decades of R&D EAPs, to employ them in medical applications is far from real now, such as their performance and response, long-term stability and mechanical properties that still need further development to achieve EAPs actuators with properties that mimic the requirements of each needed application. In addition, the preparation methods of EAPs actuators are also considered a main challenge that need further investigation to improve the characteristics of EAP actuators for using them in real applications. For example, 3D printing technique for EAPs with accurate dimensional control can lead to a revolution in EAPs developments in order to apply them in future soft robots, artificial muscles, and tissue engineering applications [79, 80].”

Reviewer 3 Report
The manuscript entitled “Electroactive Polymers-Based Composites for Artificial Mus- 1
cles-Like Actuators: A Review” has been submitted by authors. Some issues to be addressed which will improve the quality of manuscript. Therefore, I recommend this work could be published after the major revision
1. The English composition requires many improvements. The authors should proofread the manuscript carefully to minimize grammatical errors.
2. Figures 5 and 6 inner text is unclear. Make it readable, please.
3. Authors should include one schematic diagram that includes key points of review articles.
4. The characterization part and the result and discussion part are not supported by enough references. It may be supported by the recent relevant references
Materials Today,10, 2007, 30-38; J Appl Biomater Funct Mater 2015; 13 (1): 1-9; Journal of Physics and Chemistry of Solids, 149, 2021, 109774

Author Response
- The English composition requires many improvements. The authors should proofread the manuscript carefully to minimize grammatical errors.
Thank you for your kind comment. Please, be informed that the English composition was proofread.
- Figures 5 and 6 inner text is unclear. Make it readable, please.
Thank you for your comment. Kindly, check the resolution for Figures 5 and 6 which were enhanced.
- Authors should include one schematic diagram that includes key points of review articles.
Thank you for your kind suggestion. Please, check the following graphical abstract:
- The characterization part and the result and discussion part are not supported by enough references. It may be supported by the recent relevant references
Materials Today,10, 2007, 30-38; J Appl Biomater Funct Mater 2015; 13 (1): 1-9; Journal of Physics and Chemistry of Solids, 149, 2021, 109774
Thank you for your kind comment. Please, be informed that all proposed references were added to the manuscript.

Round 2
Reviewer 2 Report
It is fine for publication.
Reviewer 3 Report
I recommended accepting the author's response to all comments in their current form.